# EIEN: Endoscopic Image Enhancement Network Based on Retinex Theory

**DOI:** 10.3390/s22145464

**Published:** 2022-07-21

**Authors:** Ziheng An, Chao Xu, Kai Qian, Jubao Han, Wei Tan, Dou Wang, Qianqian Fang

**Affiliations:** 1School of Integrated Circuits, Anhui University, Hefei 230601, China; p20301227@stu.ahu.edu.cn (Z.A.); p20201085@stu.ahu.edu.cn (K.Q.); p20201029@stu.ahu.edu.cn (J.H.); p20301226@stu.ahu.edu.cn (W.T.); p20301228@stu.ahu.edu.cn (D.W.); p20201087@stu.ahu.edu.cn (Q.F.); 2AnHui Engineering Laboratory of Agro-Ecological Big Data, Hefei 230601, China

**Keywords:** image enhancement, convolutional neural network, endoscopic image, retinex, self-attention mechanism

## Abstract

In recent years, deep convolutional neural network (CNN)-based image enhancement has shown outstanding performance. However, due to the problems of uneven illumination and low contrast existing in endoscopic images, the implementation of medical endoscopic image enhancement using CNN is still an exploratory and challenging task. An endoscopic image enhancement network (EIEN) based on the Retinex theory is proposed in this paper to solve these problems. The structure consists of three parts: decomposition network, illumination correction network, and reflection component enhancement algorithm. First, the decomposition network model of pre-trained Retinex-Net is retrained on the endoscopic image dataset, and then the images are decomposed into illumination and reflection components by this decomposition network. Second, the illumination components are corrected by the proposed self-attention guided multi-scale pyramid structure. The pyramid structure is used to capture the multi-scale information of the image. The self-attention mechanism is based on the imaging nature of the endoscopic image, and the inverse image of the illumination component is fused with the features of the green and blue channels of the image to be enhanced to generate a weight map that reassigns weights to the spatial dimension of the feature map, to avoid the loss of details in the process of multi-scale feature fusion and image reconstruction by the network. The reflection component enhancement is achieved by sub-channel stretching and weighted fusion, which is used to enhance the vascular information and image contrast. Finally, the enhanced illumination and reflection components are multiplied to obtain the reconstructed image. We compare the results of the proposed method with six other methods on a test set. The experimental results show that EIEN enhances the brightness and contrast of endoscopic images and highlights vascular and tissue information. At the same time, the method in this paper obtained the best results in terms of visual perception and objective evaluation.

## 1. Introduction

Medical endoscopic images suffer from serious degradation problems such as uneven illumination and low contrast due to the complexity of the internal structures of the human body and the limitations of imaging technology [1]. These drawbacks not only make it inappropriate to observe tiny vessels and lesions with insignificant early color changes but also seriously affect the physician’s diagnosis and treatment and reduce the accuracy of some auxiliary diagnostic devices. It is undoubtedly one of the most effective methods to improve the quality of endoscopic images by image enhancement methods to improve the accuracy of disease diagnosis and the safety of minimally invasive surgical operations. Therefore, designing an effective endoscopic image enhancement method that can assist physicians in detecting suspicious lesions at an early stage is important for preventing complications and deterioration of the disease. In order to improve image quality, a large number of enhancement algorithms have been proposed by early researchers. Roughly speaking, there are four types of existing methods: histogram equalization-based methods, gamma correction-based methods, Retinex-based methods, and deep learning-based methods.

Histogram equalization (HE) [2] is used to achieve contrast enhancement by varying the dynamic pixel range of an image to be approximately uniformly distributed. However, because HE does not consider the pixel relationship between images, it leads to information loss and excessive brightness enhancement problems. To solve the issues arising from global processing, Ibrahim et al. [3] proposed to process the histogram in regions and assign a new dynamic range to each partition, allowing the image to maintain the average brightness while improving the effect. The classical CLAHE [4] algorithm, in which it is considered that the image should be processed in chunks and constrained using the method of limiting the contrast, eventually achieves good enhancement results. Although these methods are simple and convenient, they do not achieve color fidelity when applied to endoscopic images.

Secondly, Gamma Correction (GC) is also a popular method of contrast enhancement in the pixel domain, which is suitable for processing images of different luminance but requires manual setting of the appropriate gamma value. For this reason, the researchers who proposed adaptive gamma correction [5] (AGC) were inspired by probabilistic and statistical inference to determine the gamma values using PDF and CDF. Subsequently, Huang et al. [6] proposed the AGCWD algorithm to fine-tune the statistical histogram by weighting the distribution function to reduce the generation of adverse effects. Although the existing adaptive gamma correction methods effectively enhance the contrast and illumination of images, they all cause distortion and local over-enhancement of images.

Other traditional algorithms are based on the Retinex theory approach, which is a model of human color perception and assumes that the observed image can be decomposed into two components: the reflectance component and the illumination component, and generally, only the illumination component needs to be processed to achieve enhancement. Earlier single-scale Retinex [7] (SSR) and multi-scale Retinex [8] (MSR) algorithms derived from Retinex were designed to recover reflected images; however, they may add noise during enhancement distorting local details and colors of the images. To solve these problems, Jobson et al. [9] designed a multi-scale Retinex with color recovery (MSRCR) based on MSR, which shows remarkable results in color fidelity and detail retention, but leads to the phenomenon of halo artifacts in images due to improper decomposition. Fu et al. [10] proposed a weighted variational model (SRIE) for simultaneous estimation of the reflectance and illumination components of the image. Then the corrected illumination component is re-added to the reflectance component to give better naturalness to the results. Subsequently, Guo et al. [11] proposed a structure-aware smoothing model to optimize the initial illumination component and obtain the enhanced image by elemental division. Wang et al. [12] combined Retinex theory and the inverse square ratio law of illumination to propose an algorithm for endoscopic image enhancement with good results in illumination correction.

In recent years, many CNN-based methods have shown outstanding performance in image denoising, image super-resolution, and low-light image enhancement. They have contributed significantly to the development of CNN in image enhancement. For example, a depth autoencoder is proposed in LLNet [13] to enhance images without over-amplifying image features. Retinex-Net [14] applied Retinex theory to CNN and designed a decomposition network that can decompose the reflection and illumination components and combine them with the illumination correction module to enhance low-light images. To free CNNs from the limitation of training on paired datasets, Jiang et al. [15] proposed an efficient unsupervised generative adversarial network (EnlightenGAN) for image enhancement, which uses information extracted from the input itself to normalize unpaired training without using real datasets for supervised learning. Subsequently, Guo et al. [16] designed a learnable curve estimation model to adjust the dynamic pixel range of the input image with the completed training curve and achieved remarkable results.

Although these existing methods achieve superior results in solving the problem of partial image degradation, they still have some drawbacks when applied to endoscopic images, as shown in Figure 1. In addition, endoscopic image datasets of the human body are difficult to collect and involve privacy issues. To solve this problem, we used a CMOS (Complementary Metal Oxide Semiconductor) image sensor dedicated to endoscopy and an endoscope system developed by Hefei Deming Electronics Co., Ltd. (Hefei, China) to capture images of various parts of the chicken. Since animal endoscopic images and human endoscopic images have the same imaging characteristics, the completed training model can be applied to human endoscopic images. Meanwhile, we propose a novel endoscopic image enhancement network (EIEN), consisting of three parts: decomposition network, illumination correction network, and reflection component enhancement algorithm.

The main contributions of this paper are as follows:

We use a migration learning approach to retrain a pre-trained Retinex-Net decomposition network model on an endoscopic image dataset to fine-tune the decomposition network weights. The generalizability of the decomposition network in endoscopic images is improved, and suitable decomposition images are provided for illumination correction and reflection component enhancement.We propose a self-attention guided multi-scale pyramid network to implement illumination correction. It can extract image features at different scales and guide the network to generate illumination components with balanced illumination and rich details through a self-attention mechanism.We combine the imaging characteristics of endoscopic images, stretch the green channel and blue channel in the reflection component, and fuse the stretched reflection component with the original reflection component by weighting to achieve image fidelity while highlighting the contrast of blood vessels and tissues.

## 2. Related Work

**Pyramid network structure:** Image pyramids are a structure that presents images in multiple resolutions and with conceptual simplicity. The pyramid structure is divided into the Gaussian pyramid and the Laplace pyramid, which are combined by two ways of image downsampling and upsampling, respectively. A typical application of them is image fusion, where the fusion process can achieve the effect of highlighting features and details. In recent years, pyramidal structures have played an essential role in convolutional neural network image enhancement. Li et al. [17] proposed an efficient luminance-aware pyramid network to extract image features from coarse to fine so that the reconstructed images are rich in details. Jiang et al. [18] used convolutional neural networks to estimate adaptive gamma weights for different scale illumination components and then fused the corrected illumination components to obtain enhanced images. Fu et al. [19] introduced both Gaussian and Laplace image pyramidal decomposition techniques into neural networks, which greatly simplified the learning problem of the networks. Inspired by this, we propose an effective multi-scale pyramid structure for correcting the illumination components. The structure extracts feature and detail information from low-resolution images. It fuses them layer by layer toward high-resolution features, allowing the network to achieve correction of illumination components with relatively few parameters and relatively shallow depth.

**Attention Mechanism:** In recent years, many attention modules have been proposed. Their development can be divided into two directions, namely, enhanced feature aggregation and channel and spatial attention combination, whose function is to suppress redundant features while emphasizing informative features. For example, Hu et al. [20] proposed a compressive excitation (SE) block that reassigns weights to each channel by analyzing the dependencies of each channel. Concomitantly, Gao et al. [21] uses two-dimensional averaging pooling for compression and reflects the relationship between channels in the form of covariance. Wang et al. [22] used nonlocal operations to compute the interaction between any two positions without considering their distance to maintain each pixel’s position relationship. Recently, Woo et al. [23] proposed incorporating an attention mechanism in both feature channel and feature space dimensions to improve the attention of the network to crucial information by means of two attention cascades. All these approaches are beneficial for high-level tasks that require accurate feature information, such as target detection and semantic segmentation, while they may have little impact on image enhancement. To avoid producing over-enhancement, Lee et al. [24] used the negative image of the luminance channel in the original image as a self-attention, assigning lower weights to the brighter regions and higher importance to the darker areas. Inspired by this, we combine the inverse image of the illumination component and the detail-rich G and B components [25] of the endoscopic image into three channels and generate a single-channel weight map by atrous convolution and sigmoid activation function to guide the illumination correction network to obtain an illumination-balanced and detail-rich illumination components.

## 3. Methodology

This section illustrates the details of the endoscopic image enhancement network (EIEN) we designed. Inspired by Retinex theory and endoscopic imaging properties, the framework of EIEN is shown in Figure 2. The left half is a decomposition network, and the right half contains three steps of illumination and reflection component enhancement and reconstruction of enhanced images. First, to obtain the appropriate illumination and reflection components, we retrain the pre-trained decomposition network model of Retinex-Net on our endoscopic image dataset using a migration learning approach to complete the fine-tuning of the decomposition network weights. Secondly, to avoid detail loss and over-enhancement during illumination correction, feature extraction and illumination component reconstruction are performed by a self-attention guided multi-scale pyramid model. The pyramid structure allows feature and detail information to be extracted at different scales. The self-attention mechanism assigns new weights to the spatial dimensions of the extracted feature maps at each scale to avoid detail loss in the process of multi-scale feature fusion and image reconstruction. The enhancement of the reflection component is to highlight detailed information such as blood vessels. Since the gain of the blue-green channel of the endoscopic image is conducive to highlighting the blood vessel information, the green and blue channels of the reflection component are respectively stretched to obtain the stretched reflection. The stretched reflection component and the original reflection component are weighted and fused to obtain the reflection component with prominent details such as blood vessels. Finally, the enhanced illumination and the reflection component are multiplied to obtain the enhanced result. In the following, we will illustrate the implementation details of the two sub-networks and the reflection component enhancement method.

### 3.1. Decomposition Network

The classical Retinex theory is the modeling of human color perception. This theory assumes that the observed image can be decomposed into two components: reflectance, and illumination, as shown in Equation (1):(1)S=R∘I
where S represents the original image, R represents the reflection component, I represents the illumination component, and ∘ represents element-wise multiplication. However, decomposing two components from one image is a typical discomfort problem. Still, convolutional neural networks can use the captured standard image and the loss with explicit constraints to guide the model to get the correct decomposition. Our decomposition network benefits from Retinex-Net [14]. In this paper, we migrate the decomposition network model of Retinex-Net into EIEN and fine-tune the decomposition model using the collected chicken dataset to make the network applicable to endoscopic images. The model is shown in the left half of Figure 2, where the first layer of convolution in the model is a 9 × 9 convolution and the rest of the convolution operations are 3 × 3 convolutions. In addition, the losses used in the network training are as follows.

1.**Invariable Reflectance Loss****:** According to the Retinex theory, it is known that the reflection components are invariant by the nature of the object itself, so the reflection components of the abnormal/normal illumination images are similar. The abnormal illumination image contains both low-light and high-light images. The decomposition network uses pairs of images as input. It imposes reflection consistency constraints between the two images to guide the optimization of the decomposition network, whose loss is calculated as shown in Equation (2):(2)$${ℒ}_{ir}=\|R_{\mathrm{abnormal}}-R_{\mathrm{normal}}{\|}_{1}$$
where $R_{\mathrm{abnormal}}$ is the reflection component of the abnormal image, $R_{\mathrm{normal}}$ is the reflection component of the standard image, $\|\cdot {\|}_{1}$ indicates the L1 norm operation, and by minimizing ${ℒ}_{ir}$, $R_{\mathrm{abnormal}}$ and $R_{\mathrm{normal}}$ are encouraged to be similar.2.**Illumination Smoothness Loss****:** Abnormal and normal image decomposition does not have the same illumination components but should highlight the structure of the image and local details while maintaining overall smoothness. The direct use of TV minimization (ToTal Variation Minimization) [26] as a loss function can cause unsatisfactory results and over-smoothing in areas with uniform pixels or significant luminance variations. Therefore, the loss is improved based on the TV to prevent over-smoothing of the illumination component. The improved TV formula adds the gradient of the reflectance component as a weight, and the original TV function is weighted with the gradient of the reflectance component to achieve adaptive adjustment of the illuminance component structure. The illuminance component smoothing loss is shown in Equation (3):(3)$${ℒ}_{is}=\sum_{i=\mathrm{abnormal},\mathrm{normal}}\|\nabla I_{i}\circ \exp\left(-\lambda_{g}\nabla R_{i}\right)\|$$
where ∇ represents the gradient (including horizontal and vertical gradient), $\lambda_{g}$ represents the gradient-aware balance coefficient, and $\exp\left(-\lambda_{g}\nabla R_{i}\right)$ relaxes the smoothing constraint at locations with more complex image structures and illumination discontinuities, where $\lambda_{g}$ is set to 10. The smaller the gradient of the reflection component (R), the larger the weight assigned, and the smaller the gradient of the illumination component (I) will become, making the illumination component smooth.3.**Reconstruction Loss:** To further constrain the decomposition of the network, we need to focus not only on the similarity between the reconstructed results of the reflection and illumination components of the image’s own decomposition and the original image. It is also necessary to pay attention to the similarity between the combined reconstruction results of the reflection component and the illuminance component of the paired image decomposition and the abnormal/normal image. Reconstruction loss is shown in Equation (4):
(4)$${ℒ}_{\mathrm{recon}}=\sum_{i=\mathrm{abnormal},\mathrm{normal}}\sum_{j=\mathrm{abnormal},\mathrm{normal}}\lambda_{ij}\|R_{i}\circ I_{j}-S_{j}{\|}_{1}$$
where S is the original image, R is the reflection component, I is the illumination component, and $\lambda_{ij}$ is the weighting factor, $\lambda_{ij}$ = 1 when i=j and $\lambda_{ij}$ = 0.1 when i≠j.

Therefore, the total loss of the decomposition network is defined as follows:(5)$${ℒ}_{Decom}={ℒ}_{\mathrm{recon}}+0.1{ℒ}_{is}+0.01{ℒ}_{ir}$$

### 3.2. Illumination Correction Network

After decomposing the network to decompose the reflection and illumination components, the next step should be illumination correction. Since we have obtained two illuminance components, one corresponds to the normal illumination endoscopic image, and the other corresponds to the abnormal illumination endoscopic image. Therefore, the design idea of our illuminance correction network is to make it similar to the illuminance component of the normal image by learning the features and detailed information of the illuminance component in the abnormal image, combined with feature fusion and attention guidance.

Specifically, the illumination correction is implemented through a self-attention-guided pyramid module, as shown in Figure 3. The pyramid module is composed of three pyramid layers, i.e., the input I_1_ is adjusted to two downsampled versions, which are 1/2 and 1/4 times of the original size, named I_2_ and I_3_, respectively. These three inputs first start feature extraction from the lowest resolution image, then multiply the extracted final feature map with the attention weight map, and finally perform channel stitching with the features extracted from the branch of the previous layer, and repeat the above operation in the branch of the original size image for image reconstruction to get the normal illumination component. Among them, due to the small resolution of the bottom two branches, they can have a high perceptual field without special convolution processing. Therefore, the lower resolution branches consist of a small number of 3 × 3 convolution layers and residual structures. The residual structure is shown in Figure 4C. In addition, to reduce the computational effort, we halve the number of channels after each feature stitching using 1 × 1 convolution. In order to ensure that not only the network receptive field can be improved, the global feature information can be effectively fused in the feature extraction process of the original size image. We not only use a small amount of 3 × 3 convolutional layer and residual structure in the feature extraction process, but also add Pyramid Pooling Module [27] (PPM) to this layer, and introduce residual structure in PPM, as shown in Figure 4B. The module uses adaptive averaging pooling operation to generate four feature maps of 1 × 1, 2 × 2, 3 × 3, and 6 × 6 sizes, compresses the number of channels of these size feature maps by 1 × 1 convolution, then uses bilinear interpolation to upsample the feature maps into the same size as the original map, and finally performs stitching on the channels so that the network extracts more feature information that is beneficial to image reconstruction.

Self-attention mechanism: Although the pyramid network model can achieve illumination correction, the image may be over-enhanced, and details may be lost. For example, regions with high pixel values in higher exposures may appear over-brightened by network correction. Areas with lower exposures may not be noticed by the network during correction and cannot achieve good correction results. One possible solution is to process the image in chunks to avoid the drawbacks of global processing, but it increases processing complexity and may also cause block effects. Instead, inspired by Lee et al. [24], we stitched the inverse image (I_f_) of the illumination component obtained from the decomposition network and the G and B channels containing rich detail information in the endoscope to get a three-channel feature map. Then, three atrous convolutions with convolution kernels of 3 × 3 and expansion rates of 1, 2, and 5 are used to process this feature map simultaneously to obtain three single-channel feature maps. Finally, the mean values of the three feature maps are processed by the sigmoid activation function to obtain the attention weight map, which is implemented as shown in Figure 4A.

In addition, the corrected illuminance component needs to be considered not only for its illuminance smoothness but also for the similarity of the reconstructed result with the normal illuminance image after elemental multiplication of the corrected illuminance component and the reflected component. Therefore, we use the combination of illuminance smoothing loss and reconstruction loss as the total loss of the illuminance correction network, where the illuminance smoothing loss (${ℒ}_{i}$) and reconstruction loss (${ℒ}_{r}$) of the illuminance correction network is shown in Equations (6) and (7):
(6)$${ℒ}_{i}=\|R_{abnormal}\circ \hat{I}-S_{abnormal}{\|}_{1}$$
(7)$${ℒ}_{i}=\|\nabla \hat{I}\circ exp\left(-\lambda_{g}\nabla R_{abnormal}\right)\|$$
where *Î* is the corrected light component. The total loss of the light component is shown in Equation (8).
(8)$${ℒ}_{ICN}={ℒ}_{r}+0.1{ℒ}_{i}$$

### 3.3. Reflection Component Enhancement

Since blood vessels are mainly distributed in the mucosal and submucosal layers, and in terms of penetration ability, the blue and green light bands are weaker than the red light bands. In terms of absorption ability by hemoglobin, the blue and green light bands are relatively more robust than the red light bands. Thus the green and blue channels of the image contain information about the blood vessels in the mucosal layer [25]. At the same time, the presence of the mucosal layer can lead to blurred imaging of blood vessels and tissues in the endoscopic image. Therefore, in addition to illumination correction, further processing of the reflection component is still needed to highlight the information of blood vessels and tissues in the endoscopic image, and the specific processing flow is shown in Figure 5.

The study found that images with a large degree of blur have a small standard deviation and require a more significant degree of contrast stretching. Therefore, the mean and standard deviation of the image are introduced, the G channel and B channel are adaptively stretched, and the R channel is kept unchanged to achieve the effect of enhancing the contrast of blood vessels and tissues. The channel stretching method is shown in Equation (9):(9)$$I^{c}{}_{out}\left(x,y\right)=\frac{I^{c}\left(x,y\right)-\mu^{c}}{\sigma^{c}}+\tau$$
where x and y represent the horizontal and vertical coordinates in the image, respectively, c represents the image channel containing G and B channels, $I^{c}\left(x,y\right)$ represents the original image channel, $I^{c}{}_{out}\left(x,y\right)$ represents the stretched image channel, μ is the image mean, σ is the normal image deviation, where τ is a controlled variable, we set it to 0.5. As shown in Figure 6, when the value of τ gradually increases, the weighted fused image becomes brighter, and the contrast becomes lower. Therefore, we set the value of τ to 0.5 in order to make it easier for the human eye to notice and judge.

Although the stretched reflection component significantly enhances vascular and tissue contrast, reconstructing the image color produces severe color distortion leading to visual discomfort. To solve this problem, we retain only part of the detailed information of the stretched reflection component and make it weighted with the original reflection component to obtain a reflection component with great image fidelity details. The formula is as follows:(10)$$R_{out}=\frac{\left(\sigma^{G}+\sigma^{B}\right)}{2}R_{c}+R$$
where R is the original reflection component, $R_{c}$ is the stretched reflection component, and $R_{out}$ is the enhanced reflection component.

### 3.4. Network Training

The experiments were conducted on a dedicated facility in our lab, which was configured with an Intel(R) Core(TM) i7-7700K CPU@4.20GHz and equipped with a GTX1080Ti graphics card with 11 GB of video memory. We implemented the model in the PyTorch [28] framework, which is accelerated by an NVIDIA GTX1080Ti GPU. All inputs are uniformly tuned to 224 × 224, the Adam [29] optimization algorithm is used to optimize the overall parameters, and the learning rate is set to $1\times 10^{-4}$. The whole network is trained in two steps. First, the decomposition network is trained, and then the illumination correction network is trained. The batch size is set to 8, and the convergence state can be achieved after 30 epochs of training. During the testing process, our network can handle images of any size. Overall, the training process involves minimizing the losses defined as follows:(11)$${ℒ}_{Total}={ℒ}_{Decom}+{ℒ}_{ICN}$$
where ${ℒ}_{Decom}$ and ${ℒ}_{ICN}$ are the losses of the decomposition network and illumination correction network, respectively, defined in Equations (5) and (8), respectively.

## 4. Experiment Results and Discussions

In this section, some experimental results are given. Specifically, we first present the dataset used for training and testing. Then, we perform a subjective and objective analysis of the proposed EIEN method and several classical image enhancement methods. Finally, we perform an ablation study to verify the effectiveness of each component in EIEN.

### 4.1. Datasets

To our knowledge, there is no human endoscopic image dataset specifically for neural network enhancement due to constraints involving privacy and difficulty of acquisition. Based on this consideration, we acquired images with a CMOS (Complementary Metal Oxide Semiconductor) image sensor dedicated to endoscopy and an endoscopy system developed by Hefei Deming Electronics Co. A total of 192 low and normal exposure images of various parts of the chicken, such as lungs, kidneys, and intestines, were captured by varying the exposure time and sensitivity (ISO) with other configurations fixed. In order to further constrain the decomposition ability of the decomposition network and prevent the drawbacks caused by lens dithering, we generate images with higher luminance by gamma-correcting the luminance channel (Y channel) of the Ycrcb color space of some normally illuminated images, where the gamma value is 0.5. The final dataset we used for training was 292 images, i.e., 146 pairs of images; part of the training set is shown in Figure 7a. In addition, to ensure the accuracy of the model EIEN method, we cooperated with hospitals such as the First Affiliated Hospital of Anhui Medical University and the Second Affiliated Hospital of Anhui Medical University to save some poor quality images from minimally invasive surgery videos using frame-by-frame preservation as the test set. The test set included 29 images acquired by different endoscopes such as colonoscopy, cystoscopy, laparoscopy, and enteroscopy. The test set is shown in Figure 7b.

### 4.2. Subjective and Objective Analysis

In this section, we compare the results of our proposed EIEN method with the results of six representative image enhancement methods, including MSRCR [9], AGCWD [6], LIME [11], Wang et al. [12], Retinex-Net [14], Zero-DCE [16]. The performance of the proposed algorithm is evaluated from both subjective and objective aspects. The results show that the proposed method can achieve better enhancement effects for different medical endoscopic images.

#### 4.2.1. Subjective Analysis

The quality of images captured by medical endoscopes seriously affects the accuracy of doctors’ diagnosis of early lesions inside the human body and the safety of operations in minimally invasive surgery, where natural color and uniform illumination are more conducive to human eye observation. Therefore, endoscopic image enhancement should adjust image brightness and contrast while preserving image naturalness and edge details, thus highlighting blood vessels, tissues, and lesions to assist physicians. As shown in Figure 8 and Figure 9, we compare the effect of the existing method and the proposed method in this paper with five sets of images. And some local cases of poor results are also framed in Figure 9.

MSRCR prevents color distortion in images by introducing a color recovery factor, but does not entirely prevent color distortion generated by multi-scale feature processing and fusion.

AGCWD enhanced endoscopic images can appear over-enhanced in localized areas, and vascular details in dark areas can be blurred. We can clearly see through the two sets of Figure 8a,b that the enhanced image becomes brighter in local areas causing visual discomfort.

LIME achieves good results in brightness correction, and dark areas in the image are visible. Still, the excessive tendency to brightness enhancement does not improve the contrast of endoscopic images well, resulting in tiny blood vessels and tissues becoming blurred, which is not conducive to the doctor’s diagnosis and operation in minimally invasive surgery.

Wang et al. showed significant improvement in both luminance enhancement and contrast enhancement, and its shortcoming may be due to the lack of universality in the treatment of illuminance components. These cases are visible in the three sets of Figure 8a–c; some dark areas are over-enhanced or uncorrected.

Retinex-Net has a significant effect on the correction of the illumination component. Still, it only denoises the reflection component, which leads to a severe decrease in the contrast of the endoscopic image and blurs the details of some blood vessels.

Zero-DCE processes each of the three channels of the image separately using curves estimated by the depth network to be in the appropriate dynamic range, which often results in color distortion in endoscopic images. As shown in Figure 8 and Figure 9, we can see significant color distortion and loss of detail, which may result from over-stretching of the channels.

From the comparison in Figure 8 and Figure 9, it can be observed that the results obtained by EIEN highlight the vascular and tissue information. At the same time, the illumination is improved, making the images more explicit. The results in Figure 9 achieve good performance in the global enhancement, but the improvement in the local dark areas is relatively slight, which may be due to the low pixel values in the very dark areas during the illumination component reconstruction resulting in poor results. In summary, the method proposed in this paper achieves satisfactory results in illumination correction and blood vessel and tissue contrast enhancement while avoiding problems such as excessive image enhancement and color distortion.

#### 4.2.2. Objective Analysis

Objective metrics efficiently evaluate the quality merits of enhanced images and quantitatively discriminate between good and poor image enhancement methods. We chose three commonly used reference evaluation metrics: PSNR (Peak Signal to Noise Ratio) [30], SSIM (Structural Similarity) [31], and GMSD (Gradient Magnitude Similarity Deviation) [32], to measure the similarity of content and structure between the enhanced image and the ground truth image. The PSNR metric is used to evaluate the image reconstruction quality, which is the logarithmic value of the mean square error (MSE) between the original image and the processed image relative to the square of the maximum value of the signal. The SSIM metric compares three sample and outcome variables (luminance, contrast, and structure) to determine how similar the improved image and the original image are. The GMSD metric first calculates the image gradient by the Prewitt operator, then calculates the gradient amplitude, and finally uses the gradient amplitude as a feature to generate an image quality prediction score with high accuracy. In general, the higher the value of PSNR, the better the quality of the reconstructed image, the higher the value of SSIM, the smaller the change in image structure, and the smaller the value of GMSD, the smaller the image distortion. In addition, we also use the reference-free evaluation index NIQE [33] (Natural Image Quality Evaluator) to evaluate the image quality, and the smaller the value, the better the performance. The NIQE metric extracts feature from the natural landscape to test on the test image, and these features are fitted to a multivariate Gaussian model. This model actually measures the difference between an image to be tested on a multivariate distribution that is constructed from a series of features extracted from a normal image.

As shown in Table 1, Table 2, Table 3 and Table 4, we present the metric results for the three image groups a, b, and c of Figure 8, and the mean values of the metrics for the 29 images in the test set. In the table, “Average” represents the average value of the index results of the 29 images in the test set. To observe the metric averages more clearly, we changed the value of PSNR to 1/10 of the original value and the value of GMSD to 10 times the original value, and presented them in Figure 10. From the average results of each index in Table 1, Table 2, Table 3 and Table 4 and Figure 8, we can see that the method in this paper presents the best results in all three indexes, NIQE, SSIM, and PSNR. Among them, since the enhanced image has an overall change in illumination and there is no normal illumination image in the test set as a reference. Therefore, it leads to lower PSNR values, but the PSNR values obtained by this method are significantly improved compared to several other methods. SSIM is an evaluation of the distortion of an image by combining three different factors: luminance, contrast, and structure. Our method maintains the overall information of the image while enhancing the luminance and contrast, thus resulting in a higher SSIM value. This indicates that the method in this paper reconstructs high-quality images while better maintaining the subject information of the images, which is essential for medical image processing. Although the results of GMSD of our method are second only to Wang et al., the results are very similar, indicating that our approach is also guaranteed in terms of image fidelity.

In addition, to further explore the advantages and disadvantages of the method. We clearly present the parameters of the three blocks of the proposed method and the running time on an image of size 600 × 400 in Table 5. As can be seen from the table, all the blocks except the illumination correction network achieve real-time processing. Therefore, we will subsequently design the lightweight illumination correction network to achieve real-time processing while ensuring the improvement of image quality, thus further enhancing the practical value.

### 4.3. Ablation Studies

This section conducts ablation studies to reveal the impact of some essential steps and components in our design of the EIEN.

**Fine-tuning of the decomposition network model:** One of the most critical steps in implementing the method in this paper is the fine-tuning of the decomposition network model for endoscopic images, using a dataset of chickens with different illumination levels. Therefore, it is necessary to compare the decomposition results of the original Retinex-Net and the fine-tuned decomposition network. The visual comparison results are shown in Figure 11, where it can be observed that the reflection component of the EIEN decomposition is more naturalistic. The illumination component is darker in the dark region and brighter in the light area, more in line with the natural illumination. More vascular details can be seen in the illumination component of the decomposition after the weight fine-tuning. This result is soundproof of the effectiveness of decomposition network fine-tuning.

**Self-attention mechanism:** The critical component in EIEN is that we designed a self-attention mechanism in the illumination correction network for the light correction of endoscopic images. Therefore, we compare the results by removing the attention mechanism with the final results, which are shown in Figure 12. We can see that the bright areas of the corrected illuminance components are over-enhanced and have blurred details when self-attention is not introduced. With the introduction of the attention network, the corrected illuminance component prevents over-enhancement and has a good detail retention function. This is because our self-attention mechanism combines the characteristics of the inverse image of the illumination component and the characteristics of the G and B channels containing rich, detailed information in the endoscope, which can correctly guide the network for correction.

**Reflection component enhancement:** The primary step for EIEN to generate images with significant contrast of blood vessels and tissues is to perform reflection component enhancement; as shown in Figure 13, we can see that the image reconstruction results without reflection component enhancement will have significant brightness correction, but the image contrast will be reduced, resulting in unclear images. As can be seen from d in Figure 13, the details of blood vessels are visible in the reconstructed image after reflection component enhancement. This is because this step stretches the G and B channels of the reflectance component and weighted fusion of the stretched reflectance component with the original reflectance component, thus highlighting the vascular information in the mucosal and submucosal layers. In addition, as can be seen from c in Figure 13, when all three channels of the reflection component are adaptively stretched. The enhancement results in significant color distortion and blurs the vascular information to the detriment of observation.

### 4.4. Discussion on the Novelty Our Proposed Method

The novelty of our proposed method lies in the following aspects. First, to our knowledge, our proposed network is a new take on endoscopic images. Specifically, we developed a new network framework based on the Retinex theory. The network consists of a decomposition network, illumination correction network, and reflection component algorithm. Among them, the attention mechanism and reflection component enhancement algorithm are designed specifically for endoscopic images. Different from other methods based on Retinex theory, we further highlight the vascular and tissue information of the image by enhancing the reflection component while correcting the illumination component. Secondly, the designed illumination correction network comprehensively utilizes the pyramid structure, residual structure, and self-attention mechanism, which reduces the complexity of the network and avoids excessive enhancement of illumination components and loss of details. Third, the reflection component enhancement algorithm is designed by combining the imaging characteristics of endoscopic images. It solves the problem of blurred vascular and tissue imaging due to the presence of the mucosal layer. The presented results show that our method is qualitatively and quantitatively superior to other methods. This indicates that our EIEN can effectively enhance endoscopic images to obtain images with outstanding details and good visual effects.

## 5. Conclusions and Future Directions

In this paper, a convolutional neural network-based endoscopic image enhancement method (EIEN) is designed by combining the imaging characteristics of endoscopic images and the advantages of deep learning methods. The method consists of three parts: a decomposition network, illumination correction network, and reflection component enhancement algorithm. Inspired by Retinex theory, the image is decomposed into an illumination component and a reflection component by the decomposition network. Then, the illuminance correction is performed by a designed self-attention guided multiscale pyramid structure, while the reflection component is enhanced and the enhanced image is reconstructed. In this paper, the method’s performance is verified on different endoscopic images, and the necessity of the steps and components in the method is analyzed. Compared with other image enhancement algorithms, experimental results show that the technique can enhance tiny blood vessels and tissue clarity while maintaining image naturalness and solving uneven illumination and low contrast in medical endoscopic images more effectively. The method will significantly improve the accuracy of doctors’ diagnosis of diseases and the safety of minimally invasive surgical operations.

The algorithm proposed in this study also has some shortcomings, and has large room for improvement. Future research will consider creating unsupervised network models using physical and generative models, and designing and introducing more appropriate image quality losses to guide network optimization.

## Figures and Tables

**Figure 1 sensors-22-05464-f001:**
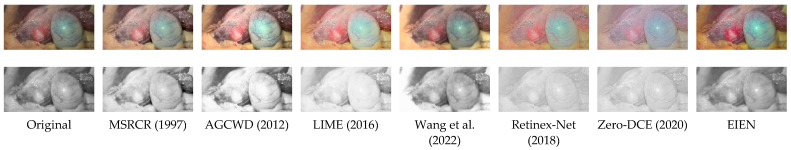
An example of image enhancement by different algorithms. The first row is the result of image enhancement, and the second row is the illumination component of the enhanced image, where the illuminance component is derived from our decomposition network. The figure shows that the enhancement results of other algorithms show over-enhancement or smoothing, while the image enhanced by EIEN has rich details [6,9,11,12,14,16].

**Figure 2 sensors-22-05464-f002:**
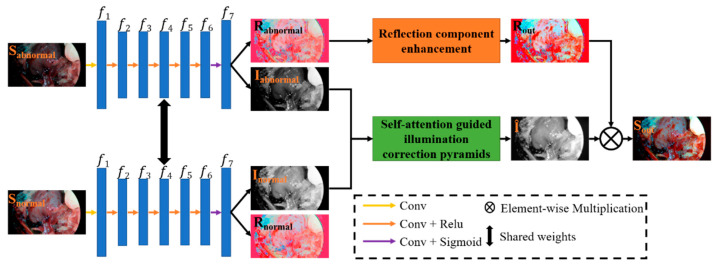
The framework of the proposed EIEN. The left half is the decomposition network, and the two images are decomposed with the same values of network weights. *f*_1_–*f*_7_ are the extracted feature maps, where *f*_1_–*f*_6_ are the features with 64 channels; *f*_7_ are the features with 4 channels, the first three channels are the reflection components, and the last channel is the illumination component. The right half is divided into three steps illumination and reflection component enhancement and reconstruction of the image.

**Figure 3 sensors-22-05464-f003:**
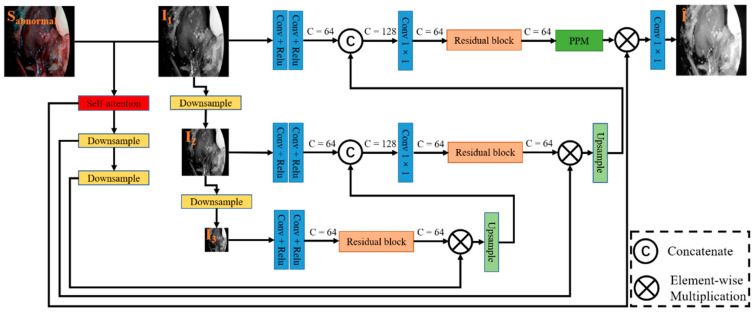
Structure of the designed illumination correction. It mainly consists of a pyramid structure, a self-attention module, a PPM module, and a residual block. The input is I_1_, the two downsampled images are I_2_ and I_3_; ‘C’ denotes the channel; downsampling is implemented using maximum pooling, where kernel size is 2 and stride is 2; the upsampling process uses bilinear interpolation.

**Figure 4 sensors-22-05464-f004:**
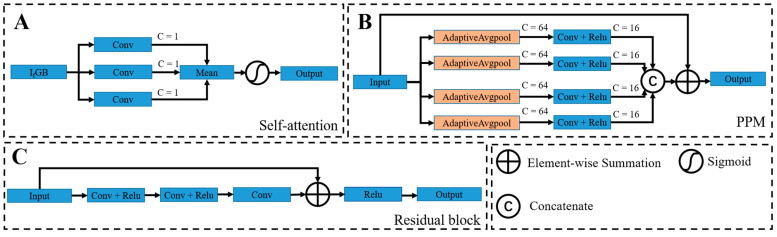
The required modules in Figure 3. It contains the self-attention mechanism (**A**), the PPM module (**B**), and the residual block (**C**). Among them, ‘C’ denotes the channel; ‘Mean’ in the blue block in A indicates the operation of taking the feature to mean; the output of PPM and residual block does not change the number of channels.

**Figure 5 sensors-22-05464-f005:**
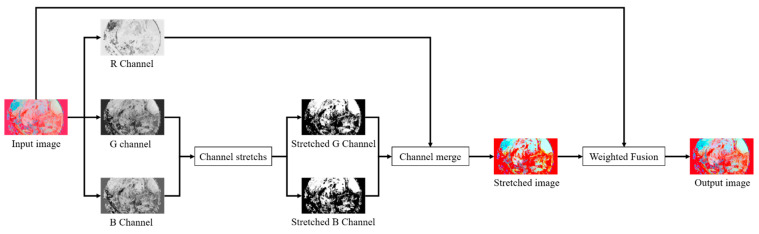
Designed method of reflective component correction.

**Figure 6 sensors-22-05464-f006:**
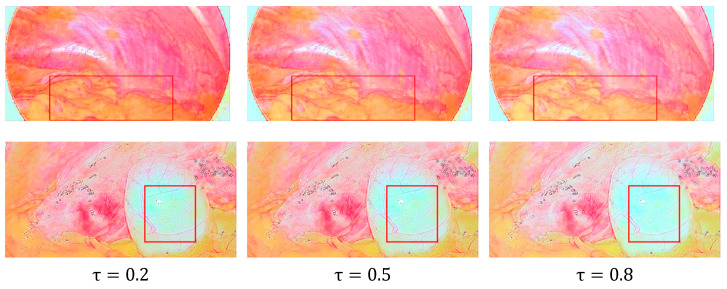
The reflection component enhancement results are obtained by setting different values of τ. Images in columns 1 and 3 suffer from visual discomfort or loss of detail, which are highlighted by red rectangles.

**Figure 7 sensors-22-05464-f007:**
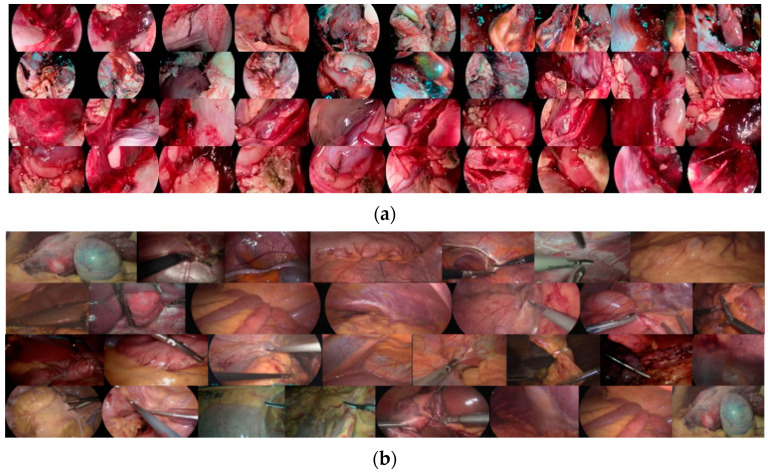
The images of the training and test sets. (**a**) shows part of the training set images; (**b**) shows the test set images.

**Figure 8 sensors-22-05464-f008:**
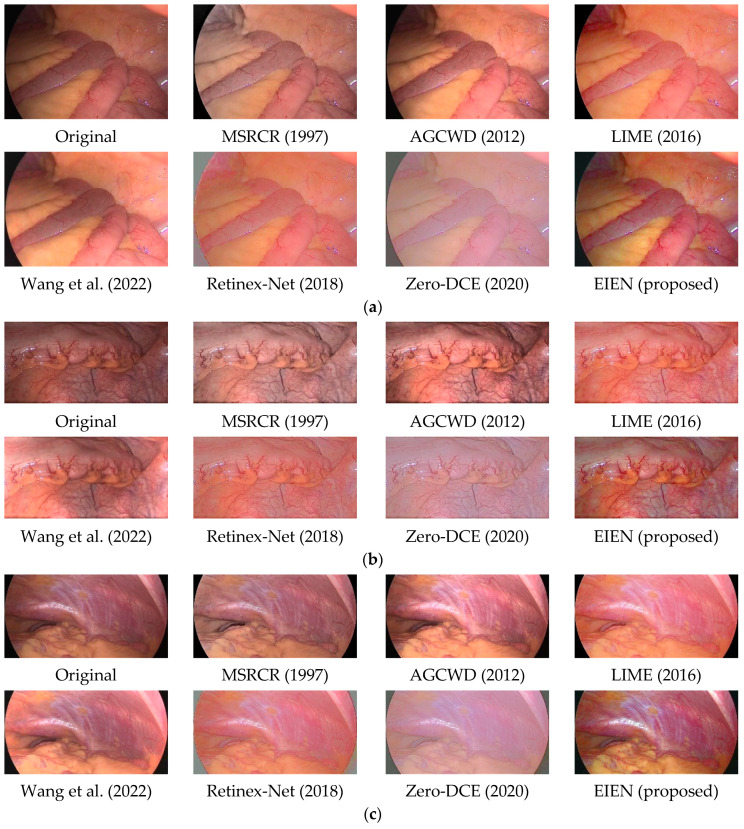
Comparison image of different methods. The three sets of images (**a**–**c**) include photos of different parts of the human body. The classical image enhancement methods and the enhancement results of the method proposed in this paper are shown [6,9,11,12,14,16].

**Figure 9 sensors-22-05464-f009:**
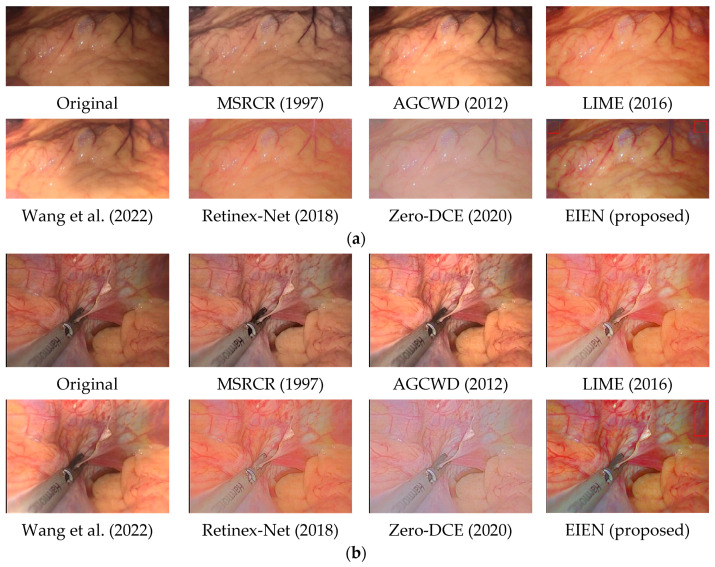
Comparison image of different methods. The two sets of images (**a**,**b**) include photos of different parts of the human body. The classical image enhancement methods and the enhancement results of the method proposed in this paper are shown. The red box is the case where the local enhancement effect is not good [6,9,11,12,14,16].

**Figure 10 sensors-22-05464-f010:**
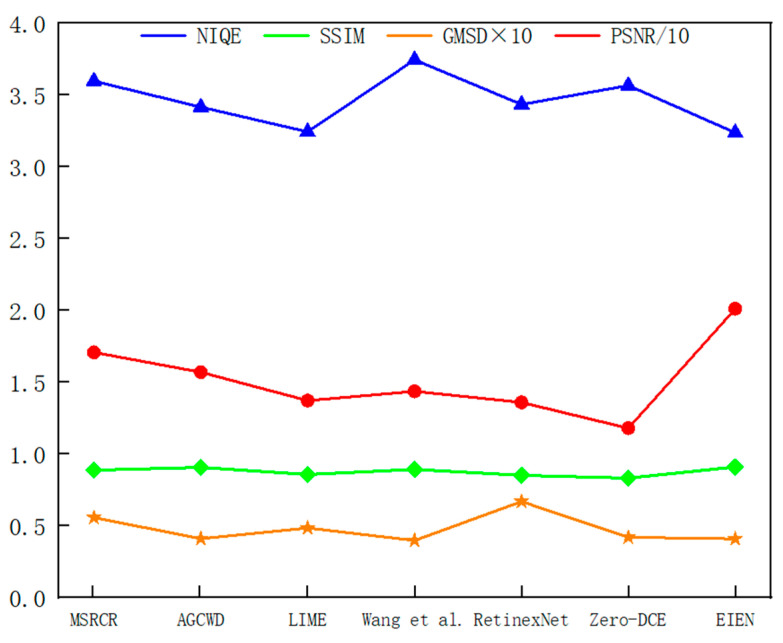
Average index comparison chart. The value of PSNR to 1/10 of the original value, and the value of GMSD to 10 times the original value.

**Figure 11 sensors-22-05464-f011:**
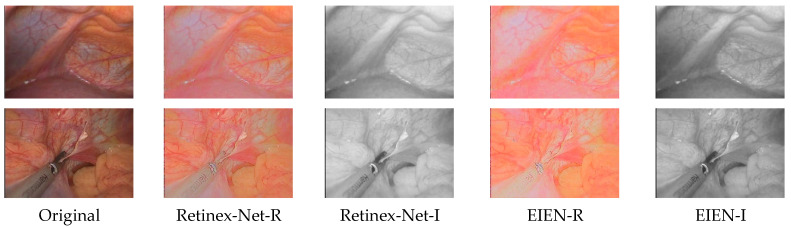
The initial decomposition results and the decomposition results after weight fine-tuning. The second and third columns show the reflectance and illuminance components of the original Retinex-Net decomposition results, respectively. The fourth and fifth columns show the reflectance and illuminance components of the decomposition after fine-tuning the weights.

**Figure 12 sensors-22-05464-f012:**
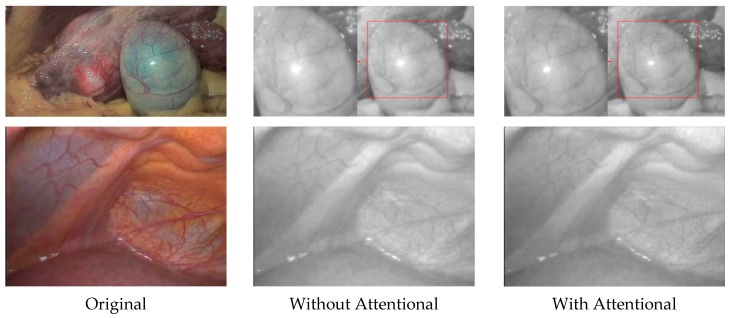
Results of comparison with and without attention mechanism. Locally over-enhanced regions are shown in the red square.

**Figure 13 sensors-22-05464-f013:**
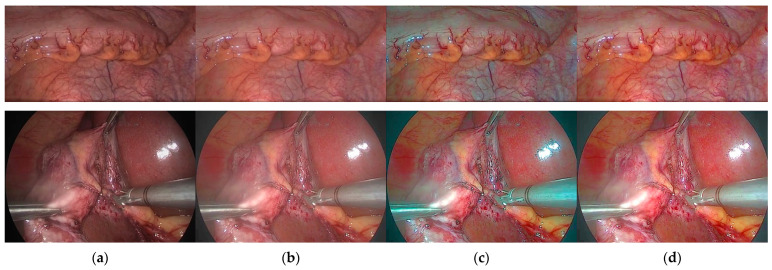
Results of comparison with and without reflection component enhancement. (**a**) Original image; (**b**) Result without reflection component enhancement; (**c**) Result of stretching all three channels of the reflective component; (**d**) G and B channels of the reflectance component are adaptively stretched, and the R channel remains unchanged.

**Table 1 sensors-22-05464-t001:** NIQE results for different algorithms. The best result is in red whereas the second best one is in blue.

Methods	Images			
	Figure 8a	Figure 8b	Figure 8c	Average
**MSRCR [9]**	3.439	3.954	3.489	3.598
**AGCWD [6]**	3.372	3.652	3.547	3.416
**LIME [11]**	2.892	3.836	2.804	3.244
**Wang et al. [12]**	3.882	3.848	4.360	3.746
**Retinex-Net [14]**	2.928	4.180	3.061	3.434
**Zero-DCE [16]**	3.353	3.905	3.613	3.567
**EIEN**	3.065	3.577	2.984	3.237

**Table 2 sensors-22-05464-t002:** SSIM results for different algorithms. The best result is in red whereas the second best one is in blue.

Methods	Images			
	Figure 8a	Figure 8b	Figure 8c	Average
**MSRCR [9]**	0.884	0.870	0.918	0.888
**AGCWD [6]**	0.941	0.870	0.942	0.908
**LIME [11]**	0.870	0.866	0.873	0.857
**Wang et al. [12]**	0.896	0.887	0.906	0.894
**Retinex-Net [14]**	0.866	0.884	0.888	0.852
**Zero-DCE** [16]	0.832	0.852	0.858	0.832
**EIEN**	0.949	0.908	0.967	0.910

**Table 3 sensors-22-05464-t003:** GMSD results for different algorithms. The best result is in red whereas the second best one is in blue.

Methods	Images			
	Figure 8a	Figure 8b	Figure 8c	Average
**MSRCR [9]**	0.058	0.048	0.042	0.056
**AGCWD [6]**	0.021	0.061	0.030	0.041
**LIME [11]**	0.042	0.036	0.045	0.049
**Wang et al. [12]**	0.030	0.043	0.037	0.040
**Retinex-Net [14]**	0.060	0.040	0.067	0.067
**Zero-DCE [16]**	0.040	0.031	0.049	0.042
**EIEN**	0.031	0.040	0.028	0.041

**Table 4 sensors-22-05464-t004:** PSNR results for different algorithms. The best result is in red whereas the second best one is in blue.

Methods	Images			
	Figure 8a	Figure 8b	Figure 8c	Average
**MSRCR [9]**	15.684	14.182	20.026	17.086
**AGCWD [6]**	16.436	15.803	16.444	15.706
**LIME [11]**	14.516	13.445	14.321	13.723
**Wang et al. [12]**	13.846	13.29	14.561	14.365
**Retinex-Net [14]**	13.547	14.265	14.102	13.595
**Zero-DCE [16]**	11.741	11.626	12.297	11.801
**EIEN**	21.428	19.648	24.543	20.112

**Table 5 sensors-22-05464-t005:** The size and running time of each module. The test was performed on GPU. EIEN-Decom stands for decomposition network, EIEN-ICN stands for illumination correction network, and EIEN-RCE stands for reflection component enhancement method.

Methods	EIEN-Decom	EIEN-ICN	EIEN-RCE	EIEN
**SIZE (M)**	0.831	1.074	-	1.905
**TIME (S)**	0.015	0.033	0.005	0.053

## Data Availability

Not applicable.

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
