# Peer review of "EIEN: Endoscopic Image Enhancement Network Based on Retinex Theory"

_sensors, 2022, doi:10.3390/s22145464_

Round 1

Reviewer 1 Report

1.       There are some typos in the abstract section.

2.       In the abstract section, please clearly highlight the proposed novelty. The way the text is written is a bit ambiguous, the connections between the proposed concepts is not well done.

3.       Please rerevise the last sentence in the abstract section.

4.       Therefore, endoscopic image enhancement has essential relevance in computer vision.” The previous sentences do not build-up to this conclusion, maybe in medical imaging (?!).

5.       The researchers found that enhancing endoscopic images by improving hardware devices is expensive to produce and […]” again it’s not clear how the researcher can check this. They usually don’t care about the expenses; the companies are more interested in that. The proposed motivation is quite weak, please rephrase.

6.       To solve this problem, we invited a professional physician to use an endoscopic image sensor to capture images of the chicken cavity with dark and normal light.” Chicken cavity?  Is that correct? Please add an argumentation on why this was selected. Moreover, the sentence could have been formulated much better. Only the professional physician can use the sensor?

7.       We migrated the decomposition network mode […]” I’m not sure what exactly this means.

8.       It’s not clear how contribution 1 is achieved.

9.       It’s not clear why Section 2 only focusses on pyramid network structure and attention mechanism. Moreover, the attention mechanist is a very hot topic and there are many, many papers published. How is the proposed approach different than the state-of-the-art approaches?

10.   Please define “perfachievedstretching”. It would be nice to proofread the manuscript before submission.

11.   Please discuss about the network design, which is quite simple …

12.   Please provide an extended discussion regarding Figure 2.

13.   The description is lacking details regarding why this procedure Is introduced like this.

14.   The “abnormal” and “normal” notations are not practical.

15.   Figure 3, how is the down sampling performed? The figure is baldly designed, hard to understand and read. Not all the components are defined. What is the difference between the horizontal and vertical rectangles? What are the labels “6” and “4”.

16.   “I1”, “I2”, “I3”, quite unprofessional notations. Where were they introduced?

17.   Figure 4, I think adding some examples in the figure might help the reader understand much better the process. It must be placed on the entire page, please check the guidance provided by the journal.

18.   The neural network was experimented with […]”, “The training process will end at 30 epochs.” The manuscript must be polished by a more experienced author. Such expression shows a lack of experience.

19.   There are some specific punctuation marks that must be added after each equation.

20.   Please clarify what exactly is the proposed novelty. There are many questions regarding the network design. Why is such a basic design proposed? There are many advances that were proposed in the last decade that were not considered.

21.   “[…] through CMOS endoscopes […]” ?

22.   Please provide more details regarding the acquisition system.

23.   “[…] the luminance channel (V channel) […]”? The luminance is usually denoted Y.

24.   Please redesign Figure 6. It shouldn’t span on multiple pages.

25.   Please add citations to the state-of-the-art methods.

26.   Figure 7, I’m not sure if the PSNR results are correct. Also, in Table 1, the PASNR values are quite small. One may conclude that the methods are equally bad.

27.   Table 1 is hard to read and understand. Image a, b, c ? Only three images? The table provides ambiguous information. Does the proposed method provide better results? The tables do not show that.

General comments:

A.      What is the proposed novelty?

B.      The proposed network design is quite basic. There are many details missing. The notation in the manuscript must be improved.

C.      The results are not convincing.

D.      In general, the manuscript is of poor quality. Many sections must be further polished/rephrased.

Reviewer 2 Report

The authors propose an endoscopic image enhancement network (EIEN) that consists of a decomposition network borrowed from Retinex-Next, an illumination correction network, and a refection component enhancement algorithm specifically designed for endoscopic images. 

The paper is well structured and easy to follow. The technical contents and originality is relatively weak in the image processing domain. But given this is a new attempt for endoscopic images, I consider the novelty is enough and would recommend a major revision. Here are my comments to help improve the paper.

  • Line 426: Please add details about how the ground truth images are collected. 
  • Section 4.2.2: Please briefly introduce each objective metrics.
  • Table 1: Please highlight numbers that corresponding to the best and second best methods.
  • Table 1: Run the tests on more images and report the mean and std of all the metrics. Results reported on 3 images may be biased on specific (and maybe cherry-picked) image contents. 
  • Failure cases. Is there any situations this method is not working very well? Providing an insightful failure case analysis would be helpful for leading the direction for future research.
  • Important ablation studies are missing. Figure 4: “the G channel and B channel are adaptively stretched, and the R channel is kept unchanged to achieve the effect of enhancing 314 the contrast of blood vessels and tissues. “ —— Usually for general image enhancement, all three channels are enhanced. Providing a comparison with all-3-channels-stretched would be insightful to show why you keep R channel untouched.
  • A break down of the running time. Fig. 2. shows several blocks of the proposed method, it would be helpful for finding out the bottleneck of the system if the authors can perform a time complexity analysis for each stage.
  • Intermediate results. Fig. 3 shows some informative intermediate results. In addition to that, I would suggest providing a comparison among the corrected illumination components from different methods for visualizing and analyzing.

Reviewer 3 Report

1.      The dataset used in this manuscript should be further declared.

2.      The authors should show the evidence of permission of this study by the Ethics Committee.

3.      There are some parameters in the proposed method, such as the 0.1 in Eq (8), the 0.01 in Eq (5), the 0.5 in Eq (9) how to determine these values and how these parameters affect the results, should be given more information and experiments.

4.      It is suggested to add the complexity analysis of the proposed model.

5.      Check the values of Fig. 7, especially for PSNR.

Round 2

Reviewer 1 Report

  1. What is the point of adding a 3rd affiliation where none of the authors are affiliated?
  2. The authors did not answer my general comments:

“A.      What is the proposed novelty?

B.      The proposed network design is quite basic. There are many details missing. The notation in the manuscript must be improved.

C.      The results are not convincing.

D.      In general, the manuscript is of poor quality. Many sections must be further polished/rephrased.”

3. My comment 27 still stands. The tables do not show that the proposed method provide an improved performance. E.g., see Table 3.

4. The PSNR values are quite low, while the corresponding SSIM values are ok. Please verify the two metrics and provide an explanation in the manuscript.

Author Response

Dear Reviewer:

Thank you for the reviewers’ comments concerning our manuscript entitled “EIEN: Endoscopic Image Enhancement Network based on Retinex Theory” (ID: sensors-1780179). Those comments are valuable and helpful for revising and improving our paper and the essential guiding significance to our research. We have studied the comments carefully and have made a correction which we hope meets with approval. Revised portions are marked in red on the paper. The significant corrections in the paper and the responses to the reviewer’s comments are follows:

Point 1: What is the point of adding a 3rd affiliation where none of the authors are affiliated?

Response 1: Thanks for the reminder, the newly added author and the added third affiliation have been removed in accordance with your and the editor's comments.

Point 2: The authors did not answer my general comments:

Response 2: We hope you can accept our sincere apologies. Due to our oversight, we did not answer your general comment. Below are my responses to those four comments.

A: What is the proposed novelty?

Response A: To our knowledge, our proposed network is a new take on endoscopic images. We base it on the Retinex theory. So, the network consists of a decomposition network, a illumination correction network, and a reflection component enhancement algorithm. Among them, the attention mechanism and reflection component enhancement algorithm are specially designed for endoscopic images.

B: The proposed network design is quite basic. There are many details missing. The notation in the manuscript must be improved.

Response B: The proposed network design is explained in our answer to Comment A. We have revised Figure 2 and Figure 3 based on your comments and those of other reviewers. Especially for Figure 3, we have split it into two parts, Figure 3 and Figure 4, and redesigned the figure. We have added some details in the explanation of the figure. Also, we removed some unnecessary labels and modified the symbols.

C: The results are not convincing.

Response C: We have added intermediate results to the processing based on your comments and those of other reviewers. For example, the illumination components of the images after enhancement by each method have been added to Figure 1. The figure in Section 3.3 shows the processing of each step of the reflection component enhancement. For the presentation of the final enhancement results, we split the original Fig. 6 into Fig. 8 and Fig. 9, from which we can see that our method has better visual effects. At the same time, we also frame some defects in Figure 9 and analyze the reasons for this situation, one for ourselves and more researchers to do further research. We split Table 1 into Tables 1-4 to clearly present the metric results. Finally, the results of the three-channel stretching of the reflection component are added in Section 4.3 to complement the experiment further.

D: In general, the manuscript is of poor quality. Many sections must be further polished/rephrased.

Response D: Thank you very much for your valuable comments and those of the other reviewers. The first submitted manuscript does have many flaws, and we have improved and rewritten the article content based on your comments.

Point 3: My comment 27 still stands. The tables do not show that the proposed method provide an improved performance. E.g., see Table 3.

Response 3: Based on your comments, we have interpreted a, b, c, and Ave in the table. The explanation section is in lines 497-500 of the text. For the performance improvement in Table 3, the mean value of the metric results we obtained in the test set differs from the value of the literature [12] by only 0.001. Moreover, the literature [12] is the latest algorithm dedicated to endoscopic image enhancement, published in 2022. It performs well in terms of GMSD. Our method achieves similar results to GMSD with other performances higher than it, indicating a great improvement of our method. And it is pointed out in lines 513-515 of the text.

Point 4: The PSNR values are quite low, while the corresponding SSIM values are ok. Please verify the two metrics and provide an explanation in the manuscript.

Response 4: Based on your comments, we selected a variety of open-source codes for the PSNR and SSIM metrics, as well as two metric codes from the Matlab and python libraries for validation. We found no problems with the tested metric values. For the low PSNR value, we found after testing and reviewing the literature that the PSNR value is lower because the enhanced image does not have the original image with normal illumination as a reference. For example, the literature (DOI:10.1109/TIP.2021.3135473.) shows that images with significant illumination correction have significantly improved image quality with PSNR values above 20. The explanation is partially added in lines 504-513 of the manuscript.

Reviewer 2 Report

The authors propose an endoscopic image enhancement network (EIEN) that consists of a decomposition network borrowed from Retinex-Next, an illumination correction network, and a refection component enhancement algorithm specifically designed for endoscopic images. 

The paper is well structured and easy to follow. 

The authors have made efforts addressing most of my comments and I recommend an AC for the updated version. 

Reviewer 3 Report

All my concerns have been addressed. I recommend this paper for publication

Round 3

Reviewer 1 Report

  1. Figure 4 is too big, it should have been designed a bit more professional. There are some blocks with a horizontal label and some with a vertical label.
  2. Please note that the purpose of each comment written by the reviewers is to help you further modify the manuscript in such a way that the issue will not occur again. E.g., for my general comment A, simply giving an answer without modifying the manuscript is not enough. Specify also why you things that the raised issue has been already solved within the manuscript.
  3. Point 2, general comment D: “The first submitted manuscript does have many flaws, and we have improved and rewritten the article content based on your comments.” This a quite a general statement. The authors need to clearly specify where the manuscript was modified and how. The authors must pay more attention to the manuscript revision process.
  4. In Tables 1-4, the authors most improve the presentation of the results. “a”, “b”, “c”, and “Ave” are ambiguous! A multi raw table head must be added on top if these columns. Moreover,  “Ave” is not a god abbreviation for average, one can use “Avg.” (or simply write Average) or “Overall”.
  5. Please pay more attention to details! Check again the entire manuscript for typos. E.g., Figure 13, line 573, there is no space between “(c)” and “results”. Moreover, since on line 572, before “(a)” there is an “.” and not an “:”, the first letter after the item labels (a), (b), …, (d), should be with a capital letter. This is quite a basic knowledge. Also, there is not always a space (“ ”) between the Figure number and the first word (e.g., Figures 2, 8, 9, and 13). Such small details show how much work was invested in preparing the manuscript, i.e., not that much in this case.
  6. Figure 6, please add an explanation in the figure’s caption regarding the red rectangles.

The manuscript must be further polished. There is still a lack of attention to details.
